# Unconventional T Cells’ Role in Cancer: Unlocking Their Hidden Potential to Guide Tumor Immunity and Therapy

**DOI:** 10.3390/cells14100720

**Published:** 2025-05-15

**Authors:** Paola Pinco, Federica Facciotti

**Affiliations:** Department of Biotechnology and Biosciences, University of Milano-Bicocca, 20126 Milan, Italy; federica.facciotti@unimib.it

**Keywords:** unconventional T cells, cancer, tumor immunology, intratumor microbiome, anticancer therapy

## Abstract

Unconventional T (UC T) cells, including invariant natural killer T (iNKT) cells, mucosal-associated invariant T (MAIT) cells, γδ T cells, and double-negative (DN) T cells, are key players in immune surveillance and response due to their properties combining innate-like and adaptive-like features. These cells are widely present in mucosal tissues, where they can rapidly respond to infections and tumor-associated changes. In fact, UC T cells can have both pro- and anti-tumoral effects, with their activity influenced by factors such as microbial composition and the tumor microenvironment. In particular, intratumoral microbiota significantly impacts the development, function, and activation of UC T cells, influencing cytokine production and shaping the immune response in various cancers. The complex crosstalk between UC T cells and the surrounding factors is discussed in this review, with a focus on how these cells might be interesting candidates to explore and exploit as anticancer therapeutic agents. However, the great potential of UC T cells, not only demonstrated in the context of adoptive cell transfer, but also enhanced through techniques of engineering, is still flanked by different challenges, like the immunosuppressive tumor microenvironment and heterogeneity of target molecules associated with some specific categories of tumors, like gastrointestinal cancers.

## 1. Introduction

The well-known and widely described panorama of conventional T cells is flanked by a second, not still completely understood and more enigmatic ground: that of unconventional (UC) T cells.

UC T cells encompass mucosal-associated invariant T (MAIT) cells, natural killer T (NKT) cells, and γδ T cells, and they have so far been significantly less characterized than conventional T cells. They typically present TCRs with less diversity compared to conventional T cells and sense a variety of molecules through their TCR, including peptides, lipids, and metabolites. UC T cells are considered innate-like T cells because they acquire effector characteristics, such as rapid cytokine release and expression of chemokine receptors and integrins, before exiting the thymus. Consequently, they accumulate in tissues sooner than conventional T cells, around the second trimester in humans and within three weeks of birth in mice [1,2,3,4,5]. In recent years, there has been a growing appreciation of the diverse role of UC T cells in cancer. Although these cells express a TCR, their innate-like properties, such as independence of antigen-mediated education via MHC molecules and rapid response to their cytokine milieu, provide these cells with the ability to quickly adapt to their surroundings. Because of their plasticity and capability to feel the environment, these cells can behave differently depending on what they sense around them. 

Moreover, together with these groups of UC T cells, a further interest in other types of unconventional T cells, like CD4 and CD8αβ, double positive and double negative ones, is currently growing and their roles in immune regulation and cytotoxicity are increasingly recognized. 

The way in which these several types of UC T cells interact with epithelial cells (ECs) and other immune cells is examined in detail in this review. The potential role of these cells in tumor development and in pro- or anti-tumoral immunity will be discussed, also providing promising avenues for their therapeutic exploitation in cancer treatment.

## 2. Unconventional T Cells: Anti- or Pro-Tumoral?

### 2.1. MAIT Cells

MAIT cells express a semi-invariant TCR with the α chain comprising TRAV1–2 TRAJ33/20/12 paired most commonly with members of the TRBV6 family and TRBV20–1 TCRβ chain in humans [6,7]. In mice, MAIT cells utilize the orthologous TRAV1-TRAJ33 TCRα chain, often paired with a TRBV19 or TRBV13 β chain [6,7]. MAIT cells recognize small metabolite products of bacterial and yeast riboflavin biosynthesis when presented by the nonpolymorphic major histocompatibility complex (MHC)-related 1 molecule (MR1). MAIT TCR especially recognizes the vitamin B2 precursors 5-(2- oxopropylideneamino)-6-D-ribitylaminouracil (5-OP-RU) and 5-(2- oxoethylideneamino)-6-D-ribitylaminouracil (5-OE-RU) presented on MR1 [8,9]. Once activated, MAIT cells produce a variety of cytokines like IFNγ, TNFα, and IL-17, together with other effector molecules, such as perforin, granulysin, and granzymes (GZM) A, GZM B, GZM K, GZM M, and GZM H [8,9]. Moreover, they can induce maturation of dendritic cells [10] and proliferation of cytotoxic CD8^+^ T cells [11]. 

Ussher et al., in 2014, demonstrated that MAIT cells can be activated also in a TCR-independent manner, directly responding to IL-12 and IL-18 stimulation, which triggers IFNγ production [12]. This suggests that these cells, in addition to classic presentation of antigens by MR1, may be activated through alternative mechanisms. As a consequence of this eclectic behavior, MAIT cells could play a role in antitumoral immunity, as demonstrated by Petley et al., where an anti-tumor function of NK cells was reported after being activated by MAIT cells-produced IFNγ [13]. In addition, MAIT cells exhibited cytolytic activity in vitro against multiple myeloma cell lines pulsed with 5-OP-RU [14] and the fact that their exhausted phenotype is associated with relapses of cancer is an additional confirmation of their potential to be cancer-fighting cells [14]. According to this, CRC-infiltrating MAIT cells have been associated with higher IL-17 production, as shown by flow cytometry, along with an increased expression of inflammatory genes such as RSG1, CCL3, and CCL4, as revealed by scRNA-seq. Due to these features, in CRC context, MAIT cells have been linked to an antitumor activity [15]. In addition, Mr1^−/−^ mice injected with B16F10 melanoma cells show reduced metastasis compared to wild type mice, reversed by adoptive transfer of MAIT cells into Mr1^−/−^ mice [16].

However, in different cancers, MAIT cells have also been associated with dysfunctional states, lacking effector function and gaining a profile similar to that of exhausted T conventional (PD-1^high^Tim-3^+^CD39^+^) [6]. For instance, in hepatocellular carcinoma (HCC), tumor-educated MAIT cells upregulate inhibitory molecules like PD-1, CTLA-4, and TIM-3, strongly decreasing IFNγ, IL17, GZM B, and perforin production [17]. In fact, high infiltration of MAIT cells in HCC significantly correlates with poor clinical outcomes [17]. In addition, a study involving ninety-nine patients with mucosal-associated cancers, such as gastric and colon cancer, levels of circulating MAIT cells were reduced because of a strong recruitment at the level of mucosal-associated cancers (MACs), mediated by CCR6-CCL20 and CXCR6-CXCL16 interactions [18]. On one side, the study underlines that circulating MAIT cells kept the potential to kill cancer cells; on the other side, the authors observed that the level of circulating MAITs was inversely correlated with N staging and carcinoembryonic antigen levels [18]. This might mean that the efficient migration of MAIT cells to MACs was followed by their exhaustion or acquisition of dysfunctional features, potentially according to another study showing a correlation between the higher tumor-infiltration by MAIT cells and the poorer CRC patients’ survival [19]. 

Thus, the dual role in cancer immunity exhibited by MAIT cells provokes a potent cytotoxicity in some tumoral contexts, but also immune suppression and poor prognosis in other types of cancer (Table 1). Further studies are needed to explain in detail which differences across tumor types may be responsible for the divergent roles played by MAIT cells, how the tumor microenvironment shapes their activity, and how these cells can be modulated to develop novel MAIT cell-based anticancer therapies. Potential solutions are discussed in the fourth section of this review.

### 2.2. NKT Cells

NKT cells are categorized into two broad classes: type 1 and type 2 NKT cells. While circulating type 2 NKT cells are more frequent in humans and express diverse TCRs conferring broader lipid antigen specificities, type 1 NKT cells, also known as invariant NKT (iNKT) cells, are less frequent, accounting for around 1% of T cells in blood and liver [46]. However, iNKT cells, extensively studied in many contexts because they are easier to detect, express a restricted T-cell receptor (TCR) and markers typically found on NK cells [47]. The TCR on iNKT cells is semi-invariant, made up of the Vα24-Jα18 chain in humans and the Vα14-Jα18 chain in mice, paired with a limited selection of β chains, which are Vβ11 in humans and Vβ8, Vβ7, and Vβ2 in mice [46,48]. This specialized TCR allows iNKT cells to recognize lipid-based antigens presented by the MHC class-I-like molecule CD1d.

The first association between iNKT cells and cancer was reported in 1995, when a strong antitumor activity was associated, upon their stimulation, with the glycolipid α-GalCer of murine sponges [49,50]. α-GalCer, the prototypic agonist of iNKT cells, in addition to its binding to the CD1d antigen presenting molecule, can synergistically act with IL-12, thus activating both iNKT and NK cells, increasing IFNγ levels and their killing activity [20]. Thus, a significant reduction of distant tumor metastasis at early stages of tumor-bearing is observed. Consistently, analyses of scRNA-Seq and imaging mass cytometry performed on tissues from pancreatic cancer liver metastasis (PCLM) mouse models showed an increased cytotoxic activity of iNKT/NK cells [21]. At the same time, an exclusion of tumor-associated macrophages from the tumor microenvironment was observed when α-GalCer treatment was administered to mice [21]. As for other UT cells, depending on environmental conditions iNKT cells can perform pro-tumorigenic activities. For example, in the context of chronic lymphocytic leukemia (CLL), iNKT cells are initially implicated in tumor surveillance and can delay the disease onset, but once the disease occurs, they become functionally impaired and their engagement, together with the expression of CD1d by Tcl1-CLL cells, highly correlate with disease progression [22]. Further evidence concerning the pro-tumor activity of iNKT cells has been found analyzing their relationship with microbiota. In colorectal cancer, iNKT cells are aberrantly activated by different tumor-associated bacteria to produce both IL-17 and Granulocyte-macrophage colony-stimulating factors (GM-CSF), consequently recruiting tumor-associated neutrophils (TANs), or chitinase 3-like-1 protein (CHI3L1), impairing cell lytic machinery [23,51].

Together, these findings underline not only the plasticity of iNKTs (Table 1), but also their great potential to be re-educated and exploited for therapeutic purposes, as suggested in the fourth section of this review.

### 2.3. γδ Cells

Like αβ T cells, γδ T cells develop in the thymus, from double negative (DN) (CD4^−^CD8^−^) cells expressing TCRs composed of γ-chains and δ-chains [52]. In humans, a Notch-independent DN pathway generates mature DN and SP (CD8+) γδ T cells, while a Notch-dependent DP pathway produces immature and rare CD4+ SP cells, followed by DP γδ T cells. Differently, in mice, strong TCR signaling without Notch signal induces γδ lineage commitment. Knowledge of the exact antigen recognition mechanism is still limited and specific ligands for γδ T cells are not well understood. The antigen recognition of γδ T cells does not seem to depend on the processing by antigen presenting cells (APCs) and consequently on the MHC-mediated presentation [53,54,55] and no general coreceptor needed have been identified so far. Depending on the local microenvironment they can differentiate into Th1-like, Th2-like, Th9-like, Th17-like, and Treg-like cells [56,57,58]. The “polyspecificity” of γδ T cells is their main feature, rendering them interesting cells with both innate and adaptive characteristics and capable of unique immune functions in different pathological and physiological conditions [59]. It has been shown that cytokines like TGF-β, IL-4 and more recently IL-21 promote the acquisition of pro-tumoral properties by both human and murine γδ T cells [60,61]. In mice, γδ T cells mediate many pro-tumor functions through expression of IL-17A, a pleiotropic cytokine that modulates the behavior of both cancer and immune cells. In particular, ApcMin/+ mice develop colorectal cancer via direct stimulation of the IL-17 receptor on intestinal epithelial cells (IECs). This is mediated by IL-17A-producing γδ T cells, which promote proliferation and inhibit CXCL9 and CXCL10 expression. Thus, anti-tumoral CD8+ T cell recruitment is prevented [24,25]. IL-17A-producing γδ T cells can also re-educate neutrophils and macrophages towards an immunosuppressive and pro-angiogenic phenotype in different cancers (i.e., breast, liver, and ovarian) [26,27,28,29].

Then, γδ T cells have been reported to potentially polarize toward FOXP3+ regulatory phenotype, suppressing the proliferation of anti-CD3/anti-CD28 stimulated peripheral blood mononuclear cells (PBMCs) [27]. This phenotype of γδ T cells accumulates in human breast cancer expressing interferon-gamma-inducible protein 10 (IP-10) [62]. Moreover, the TGF-β secreted by γδ T cells can induce the epithelial to mesenchymal transition during which tumors can escape immune detection, ultimately resulting in metastasis [30].

Despite this evidence, γδ T cells are strongly cytotoxic and can perform antitumor activity [63]. Moreover, their presence within tumors is thought to be a favorable prognostic factor, associated with better patient survival across several cancer types (myeloma, melanoma, leukemia, and bladder cancer) [31,32,33,34]. In fact, they can lyse cancer cells through different mechanisms, including the perforin-granzyme pathway [35]. For instance, in renal cancer, γδ T cells display a selective lytic potential exclusively toward autologous primary renal cancer cells, depending on the TCR and the NKG2D receptor [64]. By releasing IFNγ and cytolytic granules, γδ T cells can kill myeloma cells, recognized through ICAM-1 and mevalonate pathway metabolites on the surface [32], while the majority of melanoma-infiltrating γδ T cells is able to kill cancer cells by producing IFNγ and TNF-α [33]. In addition, in acute lymphoblastic leukemia, a higher percentage of CD8+ γδ T cells is associated with more favorable outcomes of the disease, so that these cells are now considered an early circulating biomarker of good prognosis [32].

Finally, behaving as APCs, γδ T cells process and display antigens, providing co-stimulatory signals to activate naïve *α**β* T cells [65]. Thus, in the context of gastric cancer, tumor-activated γδ T cells not only kill tumor cells efficiently, but also strongly induce primary CD4^+^ and CD8^+^
*α**β* T cells proliferation and differentiation, increasing their cytotoxic functions and abrogating immunosuppression by T regulatory cells [36] (Table 1). 

### 2.4. Others

A growing interest in minor T cells subsets, such as double negative (DN) T and double-positive (DP) T cells, has been shown in recent decades.

The transcription factors ThPOK and Runx3 regulate the differentiation of helper CD4+ and cytotoxic CD8+ T cell lineages, by, respectively, regulating the expression of the CD4 and CD8 co-receptor [37,66] on single positive (SP) T cells that enter the periphery. However, these cell fates are not mutually exclusive, because mature CD4+CD8+ double positive T cells are present in the circulation of healthy individuals and augmented in some diseases. In 2010, enhanced frequencies DP T cells in melanomas were identified. As they demonstrated broad tumor reactivity, it was hypothesized as potential participation in antitumor immune responses [38]. Then, recent studies demonstrated that DP T cells in murine and human melanoma derive from SP T cells, which through distinct transcriptional and epigenetic modulation induced via TCR signaling, just acquire the opposite co-receptor, displaying different role and functions. In mice, DP T cells showed an increased expression of Foxp3, exhibiting modest immunosuppressive features in vitro, while human DP T cells revealed a phenotype similar to that of classical cytotoxic CD8+ T cells, suggesting a potential capability of contributing to tumor eradication [37]. But, in 2019, a study revealed an association between the increased DP T cells frequency in patients with urological cancers and the polarization of CD4+ T cells toward a Th2-like phenotype, consequently bringing to a reduction of antitumor immunity and facilitating immune escape [39]. 

As for the DN T cells, they demonstrated a great cytotoxic activity towards various hematopoietic cancers (myeloma, T cell leukemia, Burkitt’s lymphoma, acute myeloid leukemia) both in vivo and in vitro without HLA restriction [40]. In particular, DN T cells isolated from the peripheral blood of healthy individuals and co-cultured with pancreatic cancer or non-small cell lung cancer cell lines in vitro, are cytotoxic and inhibit the proliferation of cancer cells [41,42]. So, even if their differentiation, regulation, and effector activities are all points that still need to be elucidated, they seem to be actively involved in both regulatory functions and in protective immune responses so far [67]. However, the subset of DN T cells is remarkably higher in patients with colorectal cancer (CRC) compared to healthy individuals and correlates with poor prognosis [43]. In addition, some studies found that TCRαβ+ DN T cells play an immunosuppressive function in mouse glioma and melanoma models and are increased in the lymph nodes of patients with advanced melanoma, conceivably promoting progression of tumor metastasis [44,45] (Table 1).

The knowledge of these two cellular subtypes is very limited so far. The influence of the tumor microenvironment in promoting the expansion of a specific phenotype of these cells has not been explored in the mentioned studies, leaving a significant gap in our understanding. In addition, a better characterization of other aspects, like the metabolic profile, for a more complete picture is required before thinking about exploiting them as therapeutic targets or tools.

## 3. Microbiome Impact on Unconventional T Cells in Cancer

In mucosal tissues, a large proportion of unconventional T cells is present and, when exposed to microbial signals, they can activate an immediate response. Since these innate-like cells play a fundamental role in immune surveillance and response, it has been proposed as a direct involvement of the microbiome in influencing their immune activities.

Microbiota exerts a role in regulating unconventional T cells starting from their development. A study conducted on mice kept in different conditions (germ-free mice, germ-free mice reconstituted with specific bacteria, and mice housed in specific pathogen-free environments) showed that they differed in TCR V7 iNKT frequencies and cytokine response to antigens [68]. This led to the conclusion that iNKT cells, exposed to different intestinal microbes and, consequently, to several types of antigens, present variable phenotypes and functions, at least in mice [68].

As demonstrated by a study conducted on both primary HCC and liver metastasis murine models [69], the microbiome dampens expression of the CXCL16 via modification of bile acid metabolism, preventing expansion of anti-tumor NKT cells [69]. In CRC, *Fusobacterium nucleatum* (*Fn*) has been recently associated with an aberrant recruiting activity of iNKT cells. iNKTs are induced to produce IL-17 and GM-CSF upon exposure to *Fn*, thus promoting tumor infiltration of neutrophils (TANs) [23]. Compared with αGalCer-primed iNKT, only *Fn*-primed iNKT cells are able to both increase the survival rate of neutrophils and to induce their recruitment [23]. Exerting a similar pro-tumor activity on iNKT cells, *Porphyromonas gingivalis* (*Pg*), an opportunistic oral pathogen previously associated with different inflammatory diseases and cancers, hampers the iNKT cell lytic machinery through increased expression of chitinase 3-like-1 protein (CHI3L1) [51]. Thus, *Pg* accelerates CRC progression in both humans and mice, impairing iNKT cytotoxic functions and promoting host tumor immune evasion.

The tissue localization of MAIT cells, similarly to what happens to iNKT cells, is critically influenced by the colonization with riboflavin-synthesizing bacteria within the first three weeks after birth [4]. Various riboflavin derivatives produced by bacteria have been identified as MAITs ligands and the structural differences in these ligands can bring a different MR1 binding and activation [70]. *Escherichia coli* (*E. coli*), *Salmonella*, *Streptococcus,* and *Mycobacterium* have been identified as bacteria recognizing and activating MAIT cells [71]. For instance, in female genital mucosa (FGT), MAIT cells stimulated with *E. coli* displayed a distinct IL-17/IL-22 profile and have an important role in the immunological homeostasis and control of microbes at this site [72]. Moreover, tumor-infiltrating MAIT cells in CRC are CD4^+^ and Foxp3^+^ and express high levels of CD39, which is an exhaustion indicator. Although *Fn* is widely correlated with CRC development and progression, as also previously stressed with the recent studies conducted on iNKTs [23], in this context it can exert a re-activating role on MAITs, in a TCR-dependent way, stimulating production of cytokines like IFNγ and TNFα [15].

The relationship between γδ T cells and microbiota is not governed simply by the anatomical site of residency or the dominant effector function of the considered γδ T cell subset. Despite the presence of the intestinal microbiota, intestinal γδ T cells can be found in similar numbers in germ-free (GF) and specific pathogen-free (SPF) mice under healthy conditions [73]. On the other hand, the numbers of liver-resident γδ17 T cells were found to have decreased in mice kept under GF conditions, compared with in their SPF counterparts [74]. In tumors, local γδ T cells can be responsible for disease onset and progression, as demonstrated by recent studies performed on lung adenocarcinoma. Stimulating IL-1β and IL-23 production by myeloid cells, the colonizing bacterial community activates lung-resident γδ T cells and provokes inflammation associated with lung cancer [75]. As proof of this, germ-free or antibiotic treated mice were significantly protected from lung cancer development induced by Kras mutation and p53 loss [75].

Overall, this growing body of evidence underscores the complex interplay between the microbiome and UC T cells, highlighting its profound impact, in both humans and mice, on immune regulation and disease progression in cancer.

## 4. Harnessing Unconventional T Cells for Cancer Treatment

The capability of UC T cells to rapidly respond to infections or tumor-associated changes via both TCR signaling and natural killer receptors (NKRs) activation makes them suitable candidates for alternative therapeutic strategies. In addition to their great plasticity and potent cytotoxic activity, they have a strongly reduced risk of inducing graft-versus-host disease (GvHD), and this implies they can be used allogeneically, solving several intrinsic problems related to adoptive cell therapy.

### 4.1. In Situ Activation and Adoptive Unconventional T Cells Transfer

The activation of NKT-mediated immune responses both through the treatment of αGC to IL-2/GM-CSF-cultured peripheral blood mononuclear cells [76] or to APCs [77], then administered intravenously to patients with advanced or recurrent NSCLC refractory to first-line chemotherapy, resulted to be well tolerated and prolonging the overall survival. In addition, a phase I clinical trial was performed of autologous in vitro expanded iNKT cells in stage IIIB–IV melanoma, bringing an evident increase in iNKT cells activity and correlated activation of other immune cells in some patients. This means that the iNKT cell therapy for advanced melanoma is feasible and can be potentially exploited or enhanced if considered also in combination with other therapies [78].

The field of adoptive MAIT cells therapy has yet to be fully explored. They are completely lacking in alloreactive potential, as demonstrated by Tourret et al. [79], and at the same time they can be easily expanded in vitro and are capable to traffic to tissues exerting potent effector functions. This constitutes a good starting point to get possible new therapeutic agents, which however would require a tight regulation due to the ubiquitous expression of MR1 and the consequent risk of a pathological over-activation of these cells [80].

γδ T cells have been tested in different clinical trials, trying to obtain an anticancer effect mediated by these cells. Attempts to expand Vδ2^+^ T cells directly in vivo, through the systemic administration of pamidronate or zoledronate in combination with IL-2, showed an objective response in few patients with prostate cancer, multiple myeloma, and renal cell cancer [81,82,83]. Moreover, in a study conducted by Wada et al., two patients out of seven showed a reduction in the number of tumor cells in ascites associated to gastric cancer after intraperitoneal injection of expanded autologous Vδ2^+^ T cells in combination with zoledronate [84]. Thus, Vδ2^+^ T cells-based immunotherapy is achievable and, overall, safe, but further studies are necessary to increase its efficacy.

DN T cells have also been studied as potential alternative immunotherapies. An example is the study by Lee et al., in 2019, where they demonstrated an in vitro donor-unrestricted cytotoxic activity of DNTs against various cancer types. It has also been observed a capability of these cells to enhance the survival of mice infused with a lethal dose of EBV-LCL and to reduce leukemia engraftment in xenograft models, making them an alternative therapeutic solution worth exploring [40].

### 4.2. CAR-Unconventional T Cells

Also engineering strategies can be applied, in fact both preclinical and clinical studies showed potent antitumor activity and wide tumor-targeting potential of CAR-iNKT cells. A recent phase I clinical trial using autologous GD2-targeting IL-15-enhanced CAR-iNKT cells to treat pediatric patients with relapsed or refractory neuroblastoma showed promising antitumor effects [85]. Moreover, given the iNKTs’ unique feature of recognizing the non-polymorphic molecule CD1d, thus preventing the induction of graft-versus-host disease (GvHD), researchers developed CAR-iNKT cells from allogeneic PBMCs. They were dually targeting CD1d to be activated and CD19^+^ chronic lymphocytic leukemia cells to exert their cytotoxic antitumor activity [86]. The high cytotoxicity against target cells and the lack of GvHD risk make also MAIT cells promising potential candidates for off-the-shelf CAR-T products. The first demonstration of the feasibility of using CAR-engineered MAIT cells against specific targets was given in 2022 by Dogan et al., who optimized a protocol of expansion and engineering of anti-CD19 and anti-Her2 CAR-MAIT cells. Respectively, these cells resulted highly efficient in killing T2, as well as Nalm6 cell lines (expressing CD19 on their surface), and Her2^+^ MDA-231 breast cancer cell line, and both of them recorded a higher cytotoxicity than the conventional CAR-T cells [87].

Moreover, experiments to generate CAR-γδ T cells were done. An example is this study by Capsomidis et al. [88], who generated GD2-targeting CAR- γδ T cells, getting an efficient expansion and an important cytotoxic activity against human neuroblastoma cell lines LAN1 and SK-N-SH. Moreover, the authors of this study hypothesized that part of the antitumor activity exerted by these cells could be due to their multifunctional nature, since they can work also as antigen-presenting cells and potentially prolong the intratumoral immune response. According to this hypothesis, the engineered γδ T cells showed capability of migrating toward tumor cells and of antigen cross presentation, confirming the potential therapeutic advantages behind these cells [88].

To optimize next-generation unconventional T cell therapies, a viable approach is to explore the synergistic potential of bispecific T cell engagers.

### 4.3. Bispecific T Cell Engagers (BiTEs)

Different from natural antibodies, BiTEs can redirect T cells to specific tumor antigens and activate T cells. Because of this, they hold great promises for cancer treatment, but face challenges due to the induction of cytokine release syndrome and the target off-tumor toxicity. Since both peripheral blood and tumor-infiltrating Vγ9Vδ2-T cells positively correlate with clinical outcome in several malignancies [89,90,91], as well as iNKT cells, which trigger a bidirectional crosstalk with APCs, thus promoting a cytotoxic lysis of CD1d^+^ tumor cells [89,92]; Lameris et al. in their studies demonstrated a successful engagement of both these cell types by using a BiTE. In particular, they projected a BiTE with transpacific properties, engaging Vγ9Vδ2-T cells and iNKTs to CD1d^+^ tumors, to trigger a strong pro-inflammatory reaction mediated by these cells, when recruited close to the tumor [93].

Similarly, a MAIT engager was developed in the context of breast cancer, to rapidly activate and allow proliferation and degranulation of MAIT cells, leading to an efficient killing of HER2^+^ cancer cells, without triggering secondary activation of other cell types like regulatory T cells and other CD4/CD8 subsets. This study shows MAIT cells as a promising approach for the treatment of solid tumors, and together with the unconventional T cells contributing to the enlargement of the therapeutic window [94].

### 4.4. Enhancing TLS via UC T Cells

An interesting, but still underexplored, field to potentially exploit for therapeutic purposes regards tumor-associated tertiary lymphoid structures (TLSs). They are organized ectopic lymphoid aggregates within the tumor microenvironment, mostly localized at the invasive margin of the tumor [95]. TLSs are crucial for the development of adaptive antitumor cellular and humoral immunity [96], mediating the activation of both T and B cells, although factors and conditions impacting the role exerted by TLSs on the immune response are still a matter of debate [97,98,99,100]. However, the correlation between the positive response to immunotherapies and the presence of these structures is demonstrated by a wealth of evidence. In particular, the predictive potential of TLSs in different solid tumors is strictly linked to B cells activity, which can target tumor cells and activate complement, and Fcγ receptor (FcγR)-mediated functions (complement-dependent cytotoxicity (CDC), antibody-dependent cell cytotoxicity (ADCC), phagocytosis, endocytosis, and the production of chemokines and cytokines) [95]. Several studies are pushing the manipulation of TLSs’ formation and function as antitumor therapy [101,102], but to our knowledge the capabilities of UC T cells and their relationship with B cells have not been fully considered, so far. He et al. [103] have recently demonstrated that α-GalCer not only stimulates iNKT cells’ migration to spleen and switch to iNKT follicular helper (iNKTfh) cells but also induces an initial and weak activation of B cells, expressing CD1d. The activation of both cellular types is completed after their cognate interaction. Similarly, another study [104] demonstrated the capacity of MAIT cells to participate in adaptive immune responses by acting as T follicular helper cells (MAITfh) within mucosal lymphoid organs. In particular, in this study Jensen et al. observed that adoptive transfer of MAIT cells into αβ T cell-deficient mice promoted B cell differentiation and increased serum *V. cholerae*-specific IgA responses. These mechanisms can be further investigated and potentially exploited to stimulate a more efficacious antitumor response. For instance, combining adoptive cell transfer and in situ activation strategies can promote UC T cells migration at the level of TLSs and increase antitumor antibody production by B cells.

However, while these strategies hold promise for enhancing antitumor immunity by leveraging the unique properties of UC T cells within TLSs, several challenges must be considered before clinical application. One major concern is the limited control over the magnitude and specificity of the immune response initiated by UC T cells, which could lead to excessive B cell activation and the production of autoreactive antibodies, increasing the risk of autoimmunity or off-target tissue damage. Additionally, the heterogeneity of TLS composition and organization across tumor types and individuals may result in unpredictable therapeutic outcomes. As such, a patient-specific strategy, guided by molecular and spatial profiling of TLSs, may be necessary to tailor UC T cell-based interventions for maximal efficacy. Moreover, careful modulation of UC T cell activation, spatial targeting to TLS-rich areas, and integration with immune checkpoint inhibition or tolerability safeguards will be essential to harness their full therapeutic potential while minimizing detrimental effects.

In conclusion, the diverse therapeutic potential of UC T cells offers promising alternatives for cancer immunotherapy. However, while advances in adoptive cell transfer, CAR-T engineering, and BiTEs have shown promising clinical results, further research is needed to optimize their efficacy and safety. Moreover, UC T cells potential role in modulating tertiary lymphoid structures opens new perspectives in enhancing antitumor humoral responses. Future efforts should focus on optimizing these strategies through combinatorial and targeted interventions, aiming for safer and more durable clinical outcomes.

## 5. Conclusions

Unconventional T cells can play both pro- and anti-tumoral roles, influenced by the environment, especially the microbial one. Microbiome impacts UC T cells by modulating their development, phenotypes, and immune responses, which can influence cancer progression and immune evasion. The plasticity of these cells makes them promising therapeutic targets or agents, and in fact they are being explored through various strategies, including adoptive transfer, in situ activation, chimeric antigen receptor (CAR)-T cell engineering, and enhancement of their tumor-specific killing activity by bispecific T cell engagers administration. All this information is summarized in Figure 1.

Despite their therapeutic potential, further research is needed to optimize UC T cell therapies. Some tumors, especially those associated with the gastrointestinal tract, showed poorly durable responses to these types of therapies, probably because of the immunosuppressive TME and the heterogeneity of target molecules. Thus, an increase in safety of UC T cells-based therapies is necessary, together with an integration of other strategies to improve tumor targeting and immune responses in cancer treatment. In addition to this, advanced neoantigen prediction and optimal UC T cells activation targets could facilitate the clinical application of these cells as therapies in GI cancers, in order to amplify the treatment landscape.

## Figures and Tables

**Figure 1 cells-14-00720-f001:**
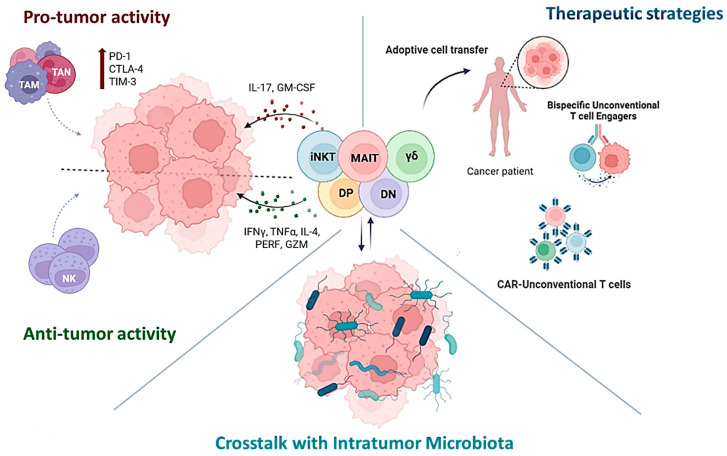
Unconventional T cells can support or fight tumors, depending on environmental factors—especially the microbiome, which shapes their development, function, and immune response. Their plasticity makes them attractive for cancer therapies, including adoptive transfer, CAR-T engineering, and bispecific T cell engagers.

**Table 1 cells-14-00720-t001:** Unconventional T cells features in tumors.

UC T Cells	Anti-Tumor Activity	Pro-Tumoral Activity	References
*MAIT*	In melanoma mouse models: IFNγ production with consequent activation of antitumor NK cells. In human CRC: IL-17 production and inflammatory genes expression. In vitro cytolytic activity against multiple myeloma cell lines.	In human HCC and MACs: dysfunctional and exhausted phenotype.	[14,15,17,18]
*NKT*	In metastatic melanoma mouse models: increased IFNγ production and killing activity after α-GalCer and IL-12 administration. In PCLM mouse models: cytotoxic activity against tumor and exclusion of TAMs from the tumor microenvironment.	In both human and murine CRC: IL-17 and GM-CSF production, with consequent recruitment of TANs; CHI3L1 production with impairment of cell lytic machinery.	[20,21,22,23]
*γδ*	In vitro IFNγ and cytolytic granules production against melanoma cells. In human gastric cancer: induction of CD4^+^ and CD8^+^ *α**β* T cells proliferation and differentiation. Better patients’ survival across several cancer types.	In CRC mouse model: IL-17 production with promotion of IECs proliferation. In breast cancer mouse model: TANs recruitment. In ovarian cancer mouse model: TAMs recruitment.	[24,25,26,27,28,29,30,31,32,33,34,35,36]
*DP*	In humans: CD8^+^-like phenotype, with a potential cytotoxic activity.	In vitro immunosuppressive activity. In human urological cancers: induction of Th2-like phenotype in CD4^+^ cells.	[37,38,39]
*DN*	In vitro cytotoxic activity.	In glioma and melanoma mouse models: immunosuppressive activity. In human CRC: poor prognosis.	[40,41,42,43,44,45]

## Data Availability

No new data were created or analyzed in this study. Data sharing is not applicable to this article.

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
