# Peer review of "Unconventional T Cells’ Role in Cancer: Unlocking Their Hidden Potential to Guide Tumor Immunity and Therapy"

_cells, 2025, doi:10.3390/cells14100720_

Round 1
Reviewer 1 Report
Comments and Suggestions for Authors
The authors provide an in-depth overview of the interactions of unconventional T cells (MAIT, NKT, γδ T cells) with tumor and immune cells, as well as insights into the therapeutic exploitation of these cells for cancer treatment. The review also includes a section dedicated to DN and DP T cells, which are subsets that are not always discussed in similar reviews. The topic is timely, especially in the context of cancer.
Major points
- Although the authors provide an up-to-date overview of current research on the subject, it would have been valuable if they had gone beyond the original conclusions of the cited studies to discuss their limitations, highlight what remains unknown or provide some hypothesis. Certain sections (i.e MAIT cells) also lack a conclusion.
- Some sentences could benefit from more precision to improve comprehension and/or avoid confusion. For instance:
Line 90-93: The authors indicate that “high infiltration of MAIT cells in HCC significantly correlate with poor clinical outcome. While, in the context of CRC, a higher IL-17 production (…) together with an increased of inflammatory genes (…) have been-associated with tumor-infiltrating MAIT”. Does it imply that IL-17 produced by MAIT cells has antitumor properties in CRC by opposition to HCC?
Line 156: “γδ T cells can reeducate neutrophils and macrophages towards an immunsuppressive and pro-angiogenic phenotype in different cancers (breast, liver and ovarian).” Is this an exhaustive list or are those just examples?
Line 184: “The transcription factors ThPOK and Runx3 regulate the differentiation of helper CD4+ and cytotoxic CD8+ T cell lineages, by epigenetically regulating the expression of either the CD4 or CD8 co-receptor”. Does it mean that ThPOK and Runx3 regulate CD4 and CD8 co-receptor respectively?
Line 177: “Finally, since they also display characteristics of APCs (…)”. Does it include phagocytosis, cross-presentation of peptides, expression of co-stimulatory molecules and secretion of polarizing cytokines?
Line 224: “A study conducted on mice kept in different conditions (germ-free mice, germ-free mice reconstituted with specific bacteria and mice housed in specific pathogen free environments) showed that they differed in TCR V7 iNKT frequencies and cytokine response to antigens”. In what way do those percentages and cytokine secretion vary depending on bacteria exposure?
Line 227: “In liver tumors (…)”. Human liver tumors? (by opposition to the previous conclusion made in mice?)
Line 82: “This suggests that the activity of MAIT cells is related to the degree of cancer progression in certain mucosal tissues”. This conclusion is somewhat vague (and/or maybe come too early in the paragraph). Does it mean that high tumor-infiltrating MAIT is associated with delayed cancer progression (in this study)? Could it be that these studies are mainly conducted at early stages of cancer, which may explain why MAIT cells are still fully functional and exert antitumor properties?
Line 131: Instead of ‘as suggested in the following paragrahs’ it may be clearer to indicate specific sections of the article (section 3 and 4), especially since the next ‘paragraph’ is not about NKT cells.
Minor points:
- Given the density of information, summarizing the pro or anti-tumor effects of each unconventional T cell subset in a table may enhance readability.
Other questions:
- Is there any information about the percentage of MAIT, γδ T , DN / DP T cells, and NKT in tumors compared to other tissues like the blood? Are there specific types of tumors in which these cells are more abundant?
Comments on the Quality of English LanguageMore concise phrasing where appropriate and improved sentence structure would enhance the overall clarity and accessibility of the review (e.g. line 50-53, line 90-93, line 125-128, line 170-173, line 177-180…)
Reviewer 2 Report
Comments and Suggestions for Authors
In the manuscript “Unconventional T cells’ role in cancer: unlocking their hidden potential to guide tumor immunity and therapy” by Pinco and Fasciotti pinpoint the importance of immune cells that, by their characteristics, do not fit the typical cell subset definitions but that may subvert and/or interfere to the function of the conventional cell subsets. Often the unconventional T cells are not taken into account in the immune response against tumours. This review reminds immunologists and oncologists of the complexity of immune system and its interplay with neoplastic cells.
The review is very well written, discuss extensively the most recent literature on the topic and, because, the various points are thoroughly organized the manuscript is easy to read. The topic is timely.
A couple of suggestions that can further improve the quality of the manuscript:
- The authors give several examples of solid tumours where unconventional T cells have been found to play a role, positive or negative. In recent years, it became also evident that the presence and the maturation of tertiary lymphoid structures in solid tumours determines disease outcome. Could the authors comment on a possible beneficial /detrimental role of unconventional T cells in the generation or maintenance of TLS? Directly or indirectly.
- Consider to draw a sort of graphical abstract to be used as main figure. This would help the reader in putting together the role of the various immune players in the context of the different tumours.
Author Response
First of all we want to thank the reviewer for the valuable advice. Below we list the changes that have been made and highlighted in yellow in the new version of the review.
Comment 1: "The authors give several examples of solid tumours where unconventional T cells have been found to play a role, positive or negative. In recent years, it became also evident that the presence and the maturation of tertiary lymphoid structures in solid tumours determines disease outcome. Could the authors comment on a possible beneficial /detrimental role of unconventional T cells in the generation or maintenance of TLS? Directly or indirectly."
Response 1: We thank the reviewer for the suggestion and we exploited the section “Harnessing unconventional T cells for cancer treatment” to discuss about the potential role of UC T cells in this structures.
Comment 2: "Consider to draw a sort of graphical abstract to be used as main figure. This would help the reader in putting together the role of the various immune players in the context of the different tumours."
Response 2: Graphical Abstract has been uploaded.
Reviewer 3 Report
Comments and Suggestions for Authors
This review article provides an informative overview of the role of different types of Unconventional T lymphocytes in cancer, highlighting their potential for anticancer therapies and discussing both strengths and limitations of utilizing and exploiting these cells in anticancer therapies. The manuscript is well-written and the focus in unconventional T cells is an interesting and emerging topic that remains underexplored in clinical settings, especially since there is a growing interest in expanding immunotherapeutic strategies beyond conventional T lymphocytes.
Some minor issues to be addressed:
1. Please add potential mechanistic insights that could improve the Unconventional T lymphocyte therapeutic approaches
2. The graphical abstract is nice, the authors could to include a Figure similar to graphical abstract in the manuscript.
3. Line 365: Please change "The plastic nature of " to "plasticity of "
4. Please check the format of the manuscript and for typographical/grammatical errors.
Author Response
First of all we want to thank the reviewer for the valuable advice. Below we list the changes that have been made and highlighted in yellow in the new version of the review.
Comment 1: "Please add potential mechanistic insights that could improve the Unconventional T lymphocyte therapeutic approaches."
Response 1: We included a paragraph in the section “Harnessing unconventional T cells for cancer treatment”, called “Enhancing TLS via UC T Cells”.
Comment 3: Line 365: Please change "The plastic nature of " to "plasticity of "
Response 3: "The plastic nature of" has been changed to "plasticity of".
Comment 4: "Please check the format of the manuscript and for typographical/grammatical errors."
Response 4: The manuscript has been completely checked.
Round 2
Reviewer 1 Report
Comments and Suggestions for Authors
The authors addressed all my comments in the revised version of the manuscript.
I have no further comments.